

# Anomaly detection for blueberry data using sparse autoencoder-support vector machine

Dianwen Wei[1], Jian Zheng[2] and Hongchun Qu[2,3]

[1] Institute of Natural Resources and Ecology, Heilongjiang Academy of Sciences, Haerbinn, China
[2] College of Computer Science and Technology, Chongqing University of Posts and Telecommunications, Chongqing, China
[3] College of Automation, Chongqing University of Posts and Telecommunications, Chongqing, China

## ABSTRACT

High-dimensional space includes many subspaces so that anomalies can be hidden in any of them, which leads to obvious difficulties in abnormality detection. Currently, most existing anomaly detection methods tend to measure distances between data points. Unfortunately, the distance between data points becomes more similar as the dimensionality of the input data increases, resulting in difficulties in differentiation between data points. As such, the high dimensionality of input data brings an obvious challenge for anomaly detection. To address this issue, this article proposes a hybrid method of combining a sparse autoencoder with a support vector machine. The principle is that by first using the proposed sparse autoencoder, the low-dimensional features of the input dataset can be captured, so as to reduce its dimensionality. Then, the support vector machine separates abnormal features from normal features in the captured low-dimensional feature space. To improve the precision of separation, a novel kernel is derived based on the Mercer theorem. Meanwhile, to prevent normal points from being mistakenly classified, the upper limit of the number of abnormal points is estimated by the *Chebyshev* theorem. Experiments on both the synthetic datasets and the UCI datasets show that the proposed method outperforms the state-of-the-art detection methods in the ability of anomaly detection. We find that the newly designed kernel can explore different sub-regions, which is able to better separate anomaly instances from the normal ones. Moreover, our results suggested that anomaly detection models suffer less negative effects from the complexity of data distribution in the space reconstructed by those layered features than in the original space.

## INTRODUCTION

An anomaly, a.k.a. an outlier, is defined as an observation that deviates so significantly from other observations as to arouse suspicion that it was generated by a different mechanism (*Chalapathy & Chawla, 2019*). As shown in Fig. 1, R1 and R2 are regions consisting of a majority of observations and considered as normal data instance regions, however, the data points in region M3, and data points P1 and P2 are few data points,

Corresponding author
Hongchun Qu, hcchyu@gmail.com

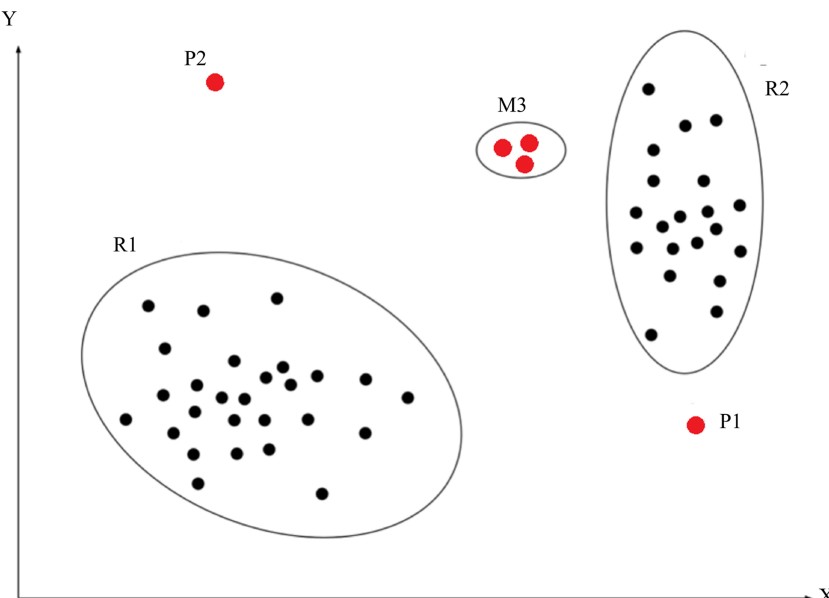

**Figure 1 Abnormal description.** The red point P1, P2 and the red points in region M3 indicate abnormal instances. These blacks points in region R1 and R2 indicate normal instances.

which are located further away from the bulk of data points, as such, these data points are considered anomalies. Anomalies can be caused by errors in the data but sometimes are also indicative of a new, previously unknown, underlying process (*Hawkins, 1980*).

Anomaly detection, a.k.a. outlier detection, is referred to as the process of detecting data instances that significantly deviate from the majority of data instances (*Pang et al., 2020*). Anomaly detection is to detect rare events and actions. However, anomalies are often irregular, for instance, one may encounter different features between anomalies, *i.e.,* differences in anomalous characteristics. Furthermore, anomalies are often rare so that it is difficult to label them. Obviously, abundance of anomaly characteristics leads to more complicated anomaly detection methods, such as hybrid detection methods consisting of deep detection methods and traditional detection methods.

It is very likely to have many subspaces in the high-dimensional space so that abnormal features can be hidden in any subspace, which brings difficulties in abnormality searching (*Aggarwal & Yu, 2001*; *Wang, Jaward Bah & Hammad, 2019*). In addition, for most anomaly detection methods, they usually measure the similarity distance between data points in order to detect anomalies. Although the measurement of distance of similarity between data points is very useful in low-dimensional spaces, it might no longer apply to the scenarios of high-dimensional spaces, since the distance of similarity between data points increase as the dimensionality of the input data increases (*Yu & Chen, 2019*), resulting in difficulty in measurement. Therefore, it is a challenge for anomaly detection in high-dimensional data.

The motivation of this study, therefore, is to achieve anomaly detection for high-dimensional data, meanwhile, to provide some insights for high-dimensional anomaly

detection. Hence, we propose a hybrid method consisting of a sparse autoencoder (SA) combined with a support vector machine, namely SA-SVM. In order to reduce the dimensionality, we use the sparse autoencoder to capture the low-dimensional features of the input data. The sparse autoencoder consists of two hidden layers, an input layer and output layer. Given that data volume and data dimension of experimental samples, two hidden layers is sufficiently large because too many hidden layers will increase the complexity of the model structure, thus increasing the training time of the model. Furthermore, we also implemented sparse items on the weights, since a sparse item offers an effective building block to learn useful features. Then, the support vector machine is used to separate abnormal features from normal features in the captured low-dimensional features. The role of the sparse autoencoder is to extract the low-dimensional features from the input data. In order to separate anomaly features in the low-dimensional features extracted by the sparse autoencoder, there is a need to derive a kernel for the SVM. Hence, the new kernel is derived. In addition, to prevent normal features from being mistakenly classified as abnormal features, the Chebyshev theorem (*Ostle & Malone, 1988*) is further used to estimate the upper limit of the number of abnormal features. Finally, the proposed method is verified and validated on both the synthetic datasets and real-world datasets.

We summarize the main contributions of this work as follows: (i) The derived kernel can explore different sub-regions, which provides better separability to differentiate anomaly instances from normal ones, so as to gain higher detection accuracy. Since the derived kernel can make the radius to be warped concave and non-decreasing, more areas with small radii can be observed. (ii) Anomaly detection models suffer less negative effects from the complexity of data distribution in the space reconstructed by those layered features than in the original space. Since the space reconstructed by those layered features provides a better spatial environment for anomaly detection.

## RELATED WORK

Most traditional anomaly detection methods are unsupervised, such as distance-based anomaly detection methods, K-nearest neighbor (KNN) (*Chehreghani, 2016*), and sampling based on rapid distance (*Sugiyama & Borgwardt, 2013*). Clearly, such methods are difficult to resist the curse of dimensionality due to relying on distance measurements. To improve the accuracy of anomaly detection, iForest (*Liu, Ting & Zhou, 2012*) defines anomalies as isolated samples to construct the isolation forests. iForest often gains better results on small scale datasets, whereas a larger number of samples may reduce the ability of iForest in outlier isolation. Because normal instances interfere with the isolation process. Moreover, iForest is more suitable for the samples in a continuous data distribution. Indeed, because of the difficulties in obtaining those deep non-linear relations between data, the traditional anomaly detection methods have high false positive rate when suffering the curse of dimensionality. In addition, including classification-based anomaly detection methods, the typical representation in such methods is support vector machine (SVM), examples such as OC-SVM (One Class-SVM) (*Erfanin et al., 2016*). Although SVM has outstanding classification ability, SVM is susceptible to the linear inseparability of high-dimensional

data, in order to compensate for this deficiency, the improved models-based SVM is designed, *e.g.*, LS-SVM (long short-SVM) (*Wang et al., 2020*).

Compared with traditional anomaly detection methods, deep network structures-based detection methods have made great success on high-dimensional data, *e.g.*, DevNet (*Pang, Shen & Hengel, 2019*), REPEN (ranking model-based framework to an Efficient method) (*Pang et al., 2018*). In addition, GAN (generative adversarial network) networks-based detection methods adopt the reconstructed error as an anomaly score, *e.g.*, AnoGAN (*Schlegl et al., 2017*). Although deep anomaly detection methods have made great success, the detection accuracy is still suboptimal since they separate feature extraction and an anomaly score, so as to only get suboptimal data representations. To gain better data representations, deep networks usually are combined with traditional anomaly detection methods, such as Deep SVDD (support vector data defintion) (*Ruff et al., 2018*), DNN-SVM (deep neural networks-support vector machine) (*Inoue et al., 2017*).

Autoencoder-based detection methods are widely used in anomaly detection. Those methods use an encoder to reduce data dimensionality as a new data representation and apply the decoder to reconstruct the input data, *e.g.*, the autoencoder proposed in *Zhou & Paffenroth (2017)*. *Slavic et al. (2022)* achieved multilevel anomaly detection through variational autoencoders and Bayesian models. Similar to *Slavic et al. (2022)* and *Li, Chang & Liu (2021)* used autoencoders for anomaly detection. Although these methods gain advanced detection results, there needs to extend the structures of autoencoders in order to improve detection accuracy. Clearly, using autoencoders encodes the input data and then anomalies can be detected in the captured low-dimensional representations (*Zhou et al., 2022*). Similarly, these examples were implemented in *Qu et al. (2021)* and *Zheng et al. (2022)*.

## METHODOLOGY

### Background

Some important lemmas and definition are given in advance to present the proposed method.

Theorem 1 (*Chen, Wang & Tsang, 2008*). Mercer theorem: when kernels are positive definite, there is one approach to obtain the mapping from original data set to feature space (*Zhang et al., 2020*). Mercer theorem indicates that any semi-positive definite symmetric function can be used as a kernel function.

Lemma 1 (*Jayasumana et al., 2014*). Let $x$ be a nonempty set. A kernel $f : (x \times x) \rightarrow \Re$ is called a positive definite kernel if $f$ is symmetric and $\sum_{i,j=1}^{n} c_i c_j f(x_i, x_j) \geq 0$ for all on $n \in N, x_1, \ldots, x_n \in \chi$ and $c_1, \ldots, c_n \in \Re$.

Lemma 2 (*Jayasumana et al., 2014*). Let $(M, d)$ be a metric space. A kernel of the form $k(x, y) = (\varphi_o d)(x, y)$, where $\varphi : \Re_0^+ \rightarrow \Re$ is a function, called a radial kernel on $(M, d)$. Furthermore, $k$ is called a continuous kernel if $\varphi$ is continuous.

Lemma 3 (*Schoenberg, 1942*). A well-known closure property of p.d (positive definite) kernels on a nonempty set, as following.

(I) If two kernels $k_1$, $k_2$ are positive definite kernels, then so is $k_1 * k_2$, and therefore $K_1^n$, for all $n \in N$.

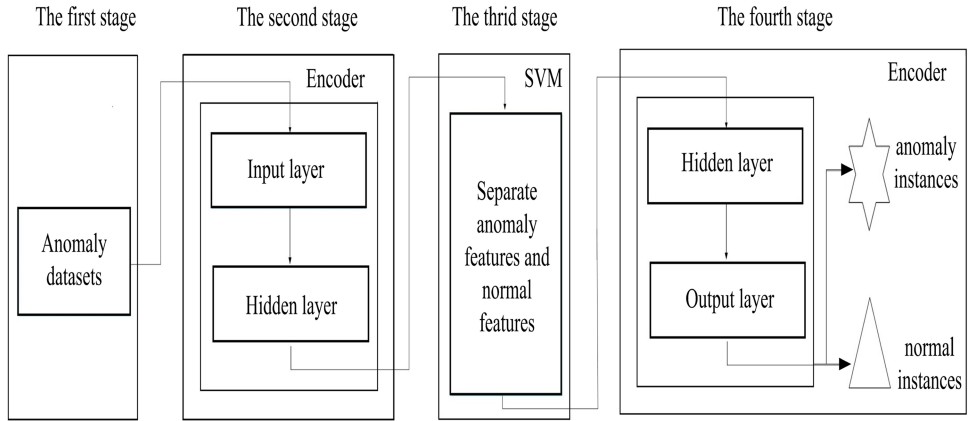

**Figure 2** The schematic description of the proposed scheme.

Lemma 4 (*Amidan, Ferryman & Cooley, 2005*). Chebyshev's inequality (otherwise known as Chebyshev's theorem (*Yu & Chen, 2019*)) was designed to determine a bound of the percentage of data that exists within $k$ number of standard deviations from the mean. For any set of observations (sample or population), the proportion of the values that lie within m standard deviations of the mean is at least 1-1/$k^2$, where $k$ is any constant greater than 1. The Chebyshev inequality is as following.

$$P(|X - \overline{u}| \geq k\sigma) \leq \frac{1}{k^2}. \tag{1}$$

where $\overline{u}$ is the data mean. $\sigma$ is the standard deviation of the data. $k$ represents the number of standard deviations from the mean.

## The proposed scheme

Figure 2 describes the proposed scheme, which consists of four stages, (i) data preparation stage; (ii) the low-dimensional feature extraction stage; (iii) the feature separation stage and (iv) the instance reconstruct stage. Firstly, the binary-classification datasets are converted into anomaly detection datasets in data preparation stage, since it is difficult to obtain the real anomaly detection datasets, we preprocess the binary-classification datasets. In the low-dimensional feature extraction stage, an encoder is used to capture the low-dimensional features from the input datasets, providing suitable spaces for anomaly discovery, and also reducing the complexity of the searching spaces. In the feature separation stage, anomaly features are separated from normal features by using the support vector machine in the space reconstructed by the captured low-dimensional features. Finally, in the instance reconstruct stage, the decoder reconstructed anomaly and normal instances based on the separated anomaly and normal features. In addition, to prevent normal instances from being misjudged as anomalies, we use the Chebyshev theorem to estimate the upper limit for the number of anomalies.

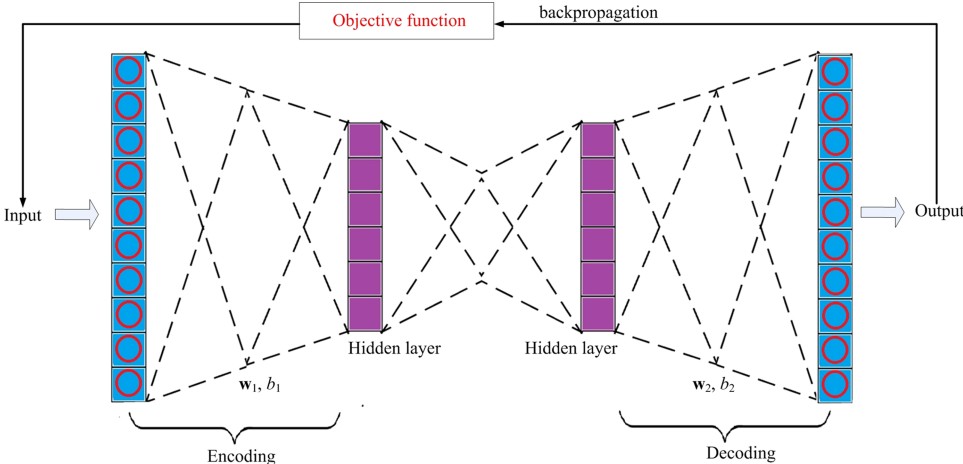

**Figure 3   The structure of the model.**

### Low-dimensional feature extraction

Given a d-dimension input sample $X^d = [x_1, x_2, x_n]^d, X^d \in \Re^{D \times N}$ and $n \geq 1$, $d>1$ is the dimensionality of the input sample, in order to achieve anomaly detection, firstly, the sparse autoencoder is used to capture the low-dimensional features $F^l = \{f_1, f_2, \ldots, f_m\}^l$ of $X^d$, $l < d$, and $m \geq 1$ is the number of low-dimensional features.

The proposed sparse autoencoder consists of two hidden layers, an input layer and an output layer, as shown in Fig. 3, of which the first hidden layer is denoted as $H_1 = [h_1^{(1)}, h_1^{(2)}, \ldots, h_1^{(m1)}]$, and the second hidden layer is denoted as, $H_2 = [h_2^{(1)}, h_2^{(2)}, \ldots, h_2^{(m2)}]$, where $H_1 \in \Re^{D \times m1}, H_2 \in \Re^{D \times m2}$ i.e., $h_1^{(j)} \in \Re^{D \times 1}, h_2^{(j)} \in \Re^{D \times 1}$ are the hidden representation of the input $x_j$, with $j \leq n$. Input layer and output layer. Input layer is used to receive the input $x_j$. The output layer sends out the corresponding reconstructed input $\hat{x}_j$.

Through using the backpropagation manner to update the objective function $J(\mathbf{w}_1, \mathbf{w}_2, b_1, b_2)$, the error between $x_j$ and the corresponding reconstructed $\hat{x}_j$ can be minimized. Therefore, $J(\mathbf{w}_1, \mathbf{w}_2, b_1, b_2)$ in the proposed sparse autoencoder is given in Eq. (2).

$$
\begin{cases}
J(\mathbf{w}_1, \mathbf{w}_2, b_1, b_2) = \dfrac{1}{n} \displaystyle\sum_{i=1}^{n} ||x_i - \hat{x}_i||^2 + \Omega_{weight} ||\mathbf{w}_1||^2 + \Omega_{weight} ||\mathbf{w}_2||^2 \\
\qquad + \displaystyle\sum_{j=1}^{D} [p * \log \dfrac{p}{\hat{p}_j} + (1-p) * \log \dfrac{(1-p)}{(1-\hat{p}_j)}] \text{(2a)} \\
\hat{p}_j = \dfrac{1}{n} \displaystyle\sum_{i=1, k=1,2}^{n} h_k^{(i)}(j) \text{(2b)}
\end{cases}
$$

where $\hat{p}_j$, $P$ are the actual activation and average activation for the $j$-$th$ neuron in the hidden layer consisting of $D$ neurons, respectively. $\mathbf{w}_1$ and $b_1$ are the weight and bias in the encoder. Similarly, $\mathbf{w}_2$ and $b_2$ are the weight and bias in the decoder. $\Omega_{weight}$ is the

sparse item for constraining the weights. For selection of $\Omega_{weight}$, we consider the empirical $\Omega_{weight}$ proposed in *Olshausen & Field (1997)*.

### Separation of abnormal features

In the reconstructed feature space, there is a certain difference between normal features and abnormal features. So we use a support vector machine (SVM) to separate the abnormal features from normal ones. Given a dataset in feature space $(\hat{x}_1, y_1), (\hat{x}_2, y_2), \ldots, (\hat{x}_N, y_N)$, where, $\hat{x}_i \in \Re^n$, $i = 1, 2, \ldots, N$, $y_i \in +1, -1$. $\hat{x}_i$ is the $i$-th eigenvector. $y_i$ is the learned class labels. The proposed SVM can be described as following,

$$
\min_{w,b,\xi_i}(\frac{1}{2}||\mathbf{w}||^2 + C\sum_{i=1}^{N}\xi_i)
$$
$$
s.t. y_i(w^T\hat{x}_i + b) \geq 1 - \xi_i
$$
$$
\xi_i \geq 0(i = 1, 2, \ldots, N)
$$
(3)

where $\xi_i$ isa slack variable. $C$ is a penalty item, and $C>0$.

This is a convex quadratic programming problem with inequality constraints, so that the dual problem can be obtained by using the Lagrange multiplier. We then transform the constrained objective function in Eq. (3) into an unconstrained newly constructed Lagrange function (*Peng & Xu, 2013*), as following

$$
L(w, b, \alpha_i) = \frac{1}{2}||w||^2 - \sum_{i=1}^{N}\alpha_i(y_i(w*\hat{x}_i + b) - 1)
$$
(4)

where $\alpha_i$ is the Lagrange multiplier, and $\alpha_i > 0$. In order to minimize $L(w, b, \alpha_i)$, let the partial derivatives of $L(w, b, \alpha_i)$ be zero with respect to $w, b$, respectively, as following

$$
\left.\begin{aligned}
\frac{\partial L(w,b,\alpha)}{\partial w} = 0 \rightarrow w = \sum_{i=1}^{N}\alpha_i y_i \hat{x}_i \\
\frac{\partial L(w,b,\alpha)}{\partial b} = 0 \rightarrow \sum_{i=1}^{N}\alpha_i y_i = 0
\end{aligned}\right\}
$$
(5)

Using Eq. (5) to solve Eq. (4), as follows

$$
\max_{\alpha}\sum_{i=1}^{N}\alpha_i - \frac{1}{2}\sum_{i=1}^{N}\sum_{j=1}^{N}\alpha_i\alpha_j y_i y_j(\hat{x}_i * \hat{x}_j)
$$
$$
s.t. \sum_{i=1}^{N}\alpha_i y_i = 0,
$$
$$
0 \leq \alpha_i \leq C, i = 1, 2, \ldots, N
$$
(6)

Equation (6) can be represented by the kernel function $\kappa(\hat{x}_i, \hat{x}_j)$, as follows

$$
\max_{\alpha}\sum_{i=1}^{N}\alpha_i - \frac{1}{2}\sum_{i=1}^{N}\sum_{j=1}^{N}\alpha_i\alpha_j y_i y_j \kappa(\hat{x}_i, \hat{x}_j)
$$
$$
s.t. \sum_{i=1}^{N}\alpha_i y_i = 0,
$$
$$
0 \leq \alpha_i \leq C, i = 1, 2, \ldots, N
$$
(7)

$\kappa(,)$ is a kernel function satisfying the Mercer theorem. Clearly, Eq. (7) has inequality constraints so the solutions must satisfy the Karush–Kuhn–Tucker (KKT) (*Peng & Xu, 2013*) conditions.

$$\left.\begin{array}{l} \alpha_i \geq 0 \\ y_i(w_i \cdot \hat{x}_i + b) - 1 \geq 0 \\ \alpha_i(y_i(w_i \cdot \hat{x}_i + b) - 1) = 0 \end{array}\right\}. \tag{8}$$

According to Eq. (5), the form of the solutions is given in Eq. (9).

$$\left.\begin{array}{l} w = \sum_{i=1}^{N} \alpha_i y_i \hat{x}_i \\ \sum_{i=1}^{N} \alpha_i y_i = 0 \end{array}\right\}. \tag{9}$$

It can be seen that in $\alpha^*$, there is at least one $\alpha_j^* > 0$ for which $j$ has

$$y_j(w^* \cdot \hat{x}_j + b^*) - 1 = 0. \tag{10}$$

Hence, the optimal weight vector $w^*$ and the optimal bias $b^*$ can be obtained as following,

$$\left.\begin{array}{l} w^* = \sum_{i=1}^{N} \alpha_i^* y_i \hat{x}_i \\ b^* = y_j - \sum_{i=1}^{N} \alpha_i^* y_i(\hat{x}_i \cdot \hat{x}_j) \end{array}\right\}. \tag{11}$$

Through learning the decision function, the separation between abnormal and normal features can be achieved in the feature space, therefore, the decision function $f(x)$ is given in Eq. (12):

$$f(x) = sign(\sum_{i=1}^{N} y_i \alpha_i^* \kappa(x_1, x_2) + b^*). \tag{12}$$

### The kernel

The kernel $\kappa(x_1, x_2)$ in Eq. (12) is used for separating abnormal features from normal features. From the Mercer theorem in the low-dimensional feature extraction section we know that $\kappa(x_1, x_2)$ is the kernel function satisfying Mercer theorem. Therefore, we use the cumulative distribution function (*Jayasumana et al., 2013*) to get a positive definite kernel

$$\kappa_f = (1 - (1 - x_1^\alpha)^\beta, 1 - (1 - x_2^\alpha)^\beta | \alpha, \beta). \tag{13}$$

where $\alpha, \beta$ are the non-negative kernel parameters.

In addition, to improve the precision of separation between abnormal and normal features, it requires a kernel to be able to perceive the location of data points, that is, the kernel needs to satisfy two properties (*Snoek et al., 2014*), *i.e.*, non-stationarity and to be

flexible in controlling searches in the normal data region. Indeed, the Matern52 kernel in *Snoek et al. (2014)* is a continuous kernel (see Lemma 2) satisfying the two properties. Because it can make the radius to be warping concave and non-decreasing (*Jayasumana et al., 2013*; *Snoek et al., 2014*), more areas with small radii can be observed. The Matern52 kernel is given in Eq. (14).

$$K_{M52} = \theta(1 + \sqrt{Cr^2(x_1, x_2)} + Ar^2(x_1, x_2))\exp{-\sqrt{Br^2(x_1, x_2)}}. \tag{14}$$

where $\theta$, $r$ are the kernel parameter and kernel radius, respectively. $A$, $B$, $C$ are constants, respectively.

Based on the above derivation, the kernel $\kappa(x_1, x_2)$ in Eq. (12) can be derived

$$\kappa(x_1, x_2) = \kappa_{M52} \circ \kappa_f. \tag{15}$$

Since kernel $\kappa_{M52}$ and kernel $\kappa_f$ are positive definite kernels, according to Lemma 3, the kernel $\kappa(x_1, x_2)$ is also a positive definite kernel. Consequently, the derivation of the kernel $\kappa(x_1, x_2)$ is completed.

### The upper boundary of the number of anomalies

When separating anomalies from the normal points, we want to count the number of anomalies in order to detect them accurately, which indeed, is difficult. However, the Chebyshev theorem (Lemma 4) can estimate the upper limit for the number of anomalous features in the feature space because it is capable of determining the upper boundary of the percentage of data that exists within $k$ number of standard deviations from the mean, meanwhile, without assuming the data distribution. Using Eq. (16), it can be estimated that the percentage of the number of anomalies is lower than $1/k^2$.

Item $C$ in Eq. (3) is a predefined penalty item, which determines the tolerated ratio that normal points are mistaken as anomalies. If $C$ is set too large, it may increase the penalty, and vice versa. For the setting of $C$, we apply Eq. (1) to estimate it, *i.e.*, let $C = P$, which can reduce the probability those normal points being mistaken for anomalies.

## Implementation

Since the proposed SA-SVM consists of a sparse autoencoder and a support vector machine, the final objective function $O(J, f)$ is composed of the objective function $J(\mathbf{w}_1, \mathbf{w}_2, b_1, b_2)$ of the sparse autoencoder in Eq. (1) and the decision function $f(x)$ of the SVM in Eq. (12)

$$O(J, f) = J(w_1, w_2, b_1, b_2) + f(x). \tag{16}$$

SA-SVM iteratively learns $O(J, f)$ until it converges, then SA-SVM outputs the detected results. The overall process of anomaly detection can be interpreted in detail as follows. (1) Firstly, the input layer of the encoder completes the mapping for $X^d$. Then, the hidden layer extracts the low-dimensional features $F^l = \{f_1, f_2, \ldots, f_m\}^l$ from $X^d$. (2) According to the extracted $F^l$, the SVM begins to perform the feature separation. Once the SVM successfully completes the linear separation between abnormal and normal features, the decoder is allowed to receive these separated features. (3) After successfully receiving the output from the SVM, the decoder reconstructs anomaly instances and normal instances,

then the output layer sends out these reconstructed instances $\hat{X}^d$. (4) Through using the backpropagation technique to learning $O(J, f)$, SA-SVM constantly minimizes the error between $X^d$ and $\hat{X}^d$, until the minimal value of $O(J, f)$ is obtained.

## SA-VM hyper-parameters and training

Training SA-SVM is to tune its hyper parameters, so we carefully studied some of hyper parameters that might affect detection.

*Learning rate.* The item is responsible for whether the objective function can converge to a local minimum. A suitable learning rate can make the objective function converge to a local minimum within a certain time. For selection of learning rate, we considered the reference value in *Qu et al. (2021)*, *i.e.,* 1e-7.

*Activation function.* We selected the sigmoid function as the activation function. Because the output of the sigmoid function is either zero or 1, it is suitable for representing anomalies and normal instances.

*The number of neurons.* According to the number of input samples, we dynamically adjust the number of neurons $\delta$ within a certain range, *i.e.,* let $\delta_1 = 20$, $\delta_2 = 100$, and $\Delta\delta = 20$, then, $\delta$ is determined using cross-validation.

*Training for SA-SVM.* The overall algorithm of SA-SVM training is given in Algorithm 1. The number of neurons $\delta$ is determined by cross-validation in step 1 and step 13. The data set $X^{Cro\_train}$ is used to train SA-SVM, then data set $X^{Cro\_val}$ is used as cross-validation of parameter $\delta$ in order to obtain the optimal configuration of $\delta$. Once the optimal configuration for $\delta$ is obtained, which is denoted as $Opt(\delta)$, the training for SA-SVM is started again, as shown from step 14 to step 19. During training, the error between the input and the reconstructed input is minimized by iteratively learning the objective function $O(J, f)$, meanwhile, the back propagation technique is used to update the hyper parameters. The training is stopped when SA-SVM converges. The process in step 20 and step 23 shows that SA-SVM is well trained then we save the trained SA-SVM, and the final training accuracy is outputted.

**Algorithm 1**. Training for SA-SVM.

Input: parameters, iteration epoch $T$, $\delta_1, \delta_2, \Delta\delta$, training set $T\_set$.

Output: training accuracy Max_TAcc.

Begin:

1 $T\_set$ is divided into $X^{Cro\_train}$, $X^{Cro\_val}$;

2 **for** $t = 1$ **to** $T$ **do**:

3   **for** $\delta = \delta_1$ **to** $_{\delta_2}$ with step $\Delta\delta$ **do**:

4     Use data set $X^{Cro\_train}$ to train SA-SVM( $X^{Cro\_train}$; $\delta$);

5     Learn objective function $O(J, f)$;

6     Use backpropagation technique to update hyper parameters until they converge;

7     Calculate training accuracy $T\_acc = $ SA-SVM ($X^{Cro\_train}$; $\delta$);

8      Use data set $X^{Cro\_val}$ to verify SA-SVM( $X^{Cro\_val}$; $\delta$);

9      Calculate validation accuracy $Val\_acc$ ($X^{Cro\_val}$; $\delta$) = SA-SVM ($X^{Cro\_val}$; $\delta$);

10      **end for**

11      Select $\delta$ so that $\delta(\max)$ = arg max($T\_acc$ ($\delta(\max)$));

12      Get the optimal for the number of neurons $Opt(\delta) = \delta(\max)$;

13 **end for**

14 **for** $t = 1$ **to** $T$ **do**:

15      Use training set $T\_set$ to train SA-SVM($T\_set$; $Opt(\delta)$);

16      Learn objective function $O(J, f)$;

17      Use backpropagation manner to update hyper parameters until they converge;

18      Calculate training accuracy $Train\_Acc$ ($t$) = SA-SVM($T\_set$; $Opt(\delta)$; $t$);

19 **end for**

20 Select the $t$ so that $t_{\max}$ = arg max ($Train\_Acc$ ($t$));

21 Get the maximum training accuracy in $t_{\max}$-th iteration $Max\_TAcc$ = SA-SVM($T\_set$; $Opt(\delta)$; $t_{\max}$);

22 Save the trained SA-SVM ($T\_set$; $Opt(\delta)$; $t_{\max}$);

23 Output the maximum training accuracy $Max\_TAcc$

     End

## EXPERIMENT SETTINGS

### Dataset

Ten synthetic datasets (*i.e.,* S1–S10) containing small proportion of potential anomalies were generated (the generation procedure follows *Campos et al., 2016*). Datasets S1, S2, and S3 were generated using regular data distributions, while dataset S4 was generated using irregular random data distribution. Datasets S1-S4 were used to verify the abnormal detection capability of different methods on different data distributions. Figure 4 shows data distribution of the four synthetic datasets. In addition, the other six synthetic datasets, *i.e.,* S5–S10 with different characteristic in data volume were used to test the running time of the proposed and comparison methods. Table 1 gives description of the ten synthetic datasets.

In addition, five high-dimensional University of California, Irvine (UCI) datasets were also used to verify the ability of anomaly detection in high-dimensional spaces. The five high-dimensional UCI datasets are often used for classification, so the procedure described in *Campos et al. (2016)* was employed to convert them into the format that is suitable for anomaly detection. The detailed description of the five UCI datasets is shown in Table 2. To eliminate randomness, 5-fold cross-validation was implemented for the proposed method and the comparison methods. We randomly divided the five UCI datasets into two parts, where one part was used as training set, including three datasets. The rest of datasets was used as testing set.

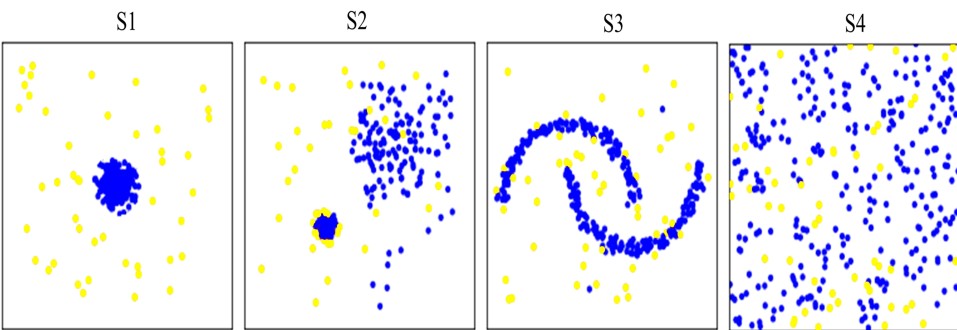

**Figure 4  Visualization of data distribution of the four synthetic datasets S1~S4.** Anomalies are marked as yellow circles. Normal points are marked as blue circles. S1 S2 and S3 are generated as regular data distributions. S4 are generated as irregular random data distribution.

**Table 1  The synthetic datasets.**

| Dataset | Abnormal ratio | Data dimensionality | Training of data volume | Testing of data volume |
|---------|----------------|---------------------|-------------------------|------------------------|
| S1 | 5% | 2 | 1,000 | 500 |
| S2 | 5% | 2 | 1,000 | 500 |
| S3 | 5% | 2 | 1,000 | 500 |
| S4 | 5% | 2 | 1,000 | 500 |
| S5 | 5% | 2 | 1,000 | 500 |
| S6 | 5% | 2 | 2,000 | 500 |
| S7 | 5% | 2 | 3,000 | 500 |
| S8 | 5% | 2 | 4,000 | 500 |
| S9 | 5% | 2 | 5,000 | 500 |
| S10 | 5% | 2 | 6,000 | 500 |

**Table 2  The UCI datasets.**

| Serial number | Datasets | Number | | Anomaly ratio | Data dimensionality |
|---------------|----------|--------|--|---------------|---------------------|
| | | Normal | Anomaly | | |
| U1 | speech | 1,023 | 17 | 1.64% | 26 |
| U2 | musk | 6,387 | 211 | 3.20% | 166 |
| U3 | mnist | 6,903 | 700 | 9.20% | 100 |
| U4 | optdigits | 5,452 | 168 | 3.00% | 64 |
| U5 | statlog | 6,358 | 77 | 1.20% | 36 |

## Comparison methods

We compared the proposed SA-SVM with three state-of-the-art detection methods, including traditional detection method iForest (*Liu, Ting & Zhou, 2012*), deep detection method DevNet (*Pang, Shen & Hengel, 2019*), and hybrid detection method REPEN (*Pang et al., 2018*), DNN-SVM (*Inoue et al., 2017*). In addition, in order to verify the ability of the derived kernel, without changing our AE-SVM structure, two benchmark models were

designed, *i.e.,* AE-SVM uses the kernel RBF, denoted as AE-SVM (RBF), and AE-SVM uses the kernel sigmoid, namely AE-SVM (sigmoid). To have fair comparisons, the optimal parameters used for the three competing methods were obtained from the corresponding literature (*Liu, Ting & Zhou, 2012*; *Slavic et al., 2022*; *Campos et al., 2016*).

### Environment settings

We implemented the proposed method and the comparison methods using Python 3.7 in Tensorflow 2.0 of Linux operating system. All experiments were run on the server with Intel i5 3.4 GHz CPU, 8G memory and. Unless otherwise state, entire experiments were run on the same GPU, using the same environment.

### Assessment metrics

We used the accuracy metric as a measurement. In addition, F1-score was also considered as an evaluation metric. The calculation formulas are as follows,

$$\text{Accuracy} = \frac{\text{TP} + \text{TN}}{\text{TP} + \text{FP} + \text{TN} + \text{FN}} \tag{17}$$

$$\text{F1} - \text{score} = \frac{2\text{TP}}{2\text{TP} + \text{FP} + \text{FN}}. \tag{18}$$

TP is the proportion of correctly predicted anomalies. TN is the proportion of correctly predicted normal instances. FP is the proportion of predicted normal instances but were anomalies. FN is the proportion of predicted anomalies but were normal instances. In addition, to test the ability of the four methods in anomaly detection, the sensitive metric was also used, as following,

$$\text{Sensitivity} = \frac{\text{TP}}{\text{TP} + \text{FN}}. \tag{19}$$

## RESULTS

### Experiments on synthetic datasets

Results on the synthetic datasets show that the proposed SA-SVM outperforms the three competitors in terms of detection performance (including the accuracy metric, F1-score metric and the sensitive metric), as shown in Fig. 5. As data distribution becomes more complicated, all methods presented a down trend in detection performance. Nevertheless, the performance of SA-SVM drops more slowly than the competitors, implying that SA-SVM has stronger anomaly detection ability on data with complex distribution. To present an intuitive comparison, Fig. 6 shows the detected results, which showed that the separated boundaries learned by SA-SVM are better than that of the three comparison methods. Together, results shown in Figs. 5–6 confirmed that the advanced detection results can be obtained in the feature space reconstructed by neural network methods.

To observe the three kernels, we visualized the change of process of the three kernels on the four synthetic datasets, as shown in Fig. 7. Figure 7A displays that the derived
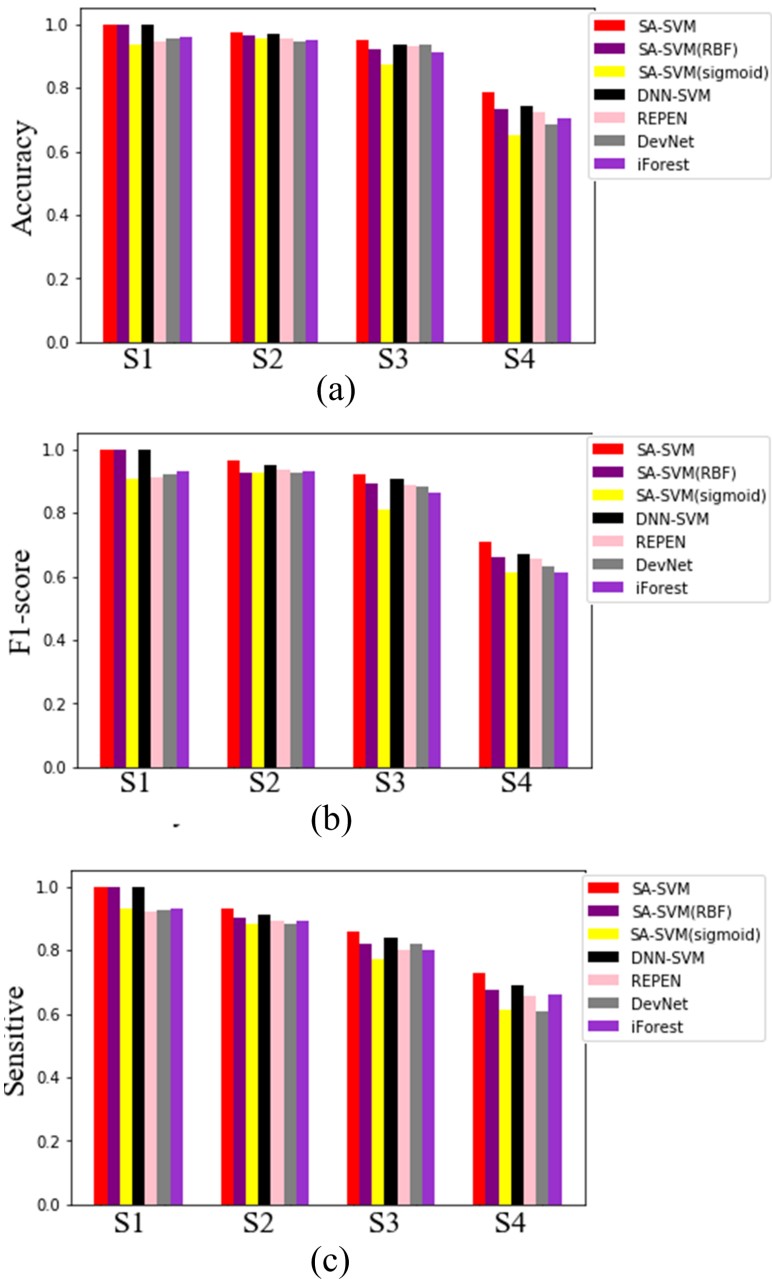

**Figure 5  Detected results on the four synthetic datasets.**

kernel can focus more on sub-regions than both the kernel RBF in Fig. 7B and the kernel sigmoid in Fig. 7C. It can be seen that as the complexity of the data distribution increases, so does the number of sub-regions that the derived kernel focuses on. Especially, on the synthetic dataset S4 generated by irregular random data distribution, the derived kernel finds more sub-regions than RBF kernel does. This mean the derived kernel provides better separability than the RBF kernel during the separation of anomaly features and abnormal features. Overall, the results indicate that the derived kernel is capable of exploring different

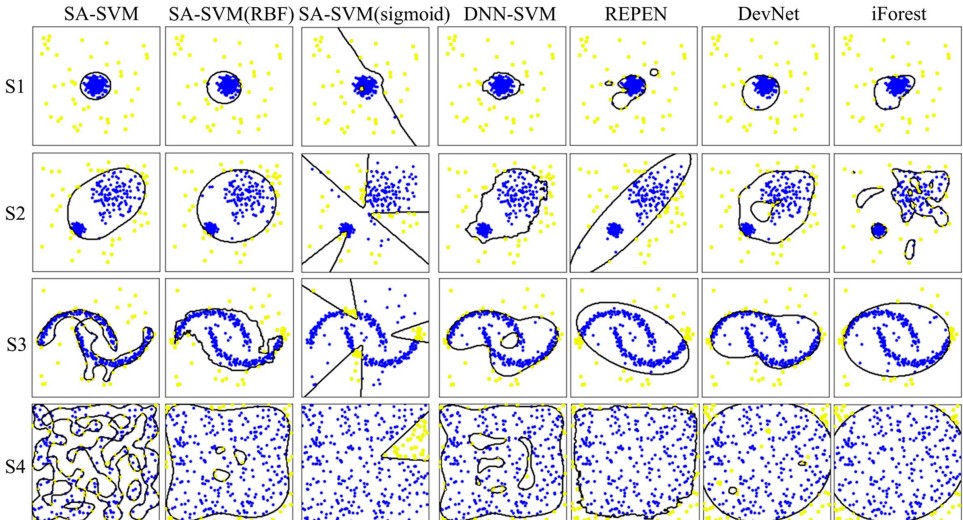

**Figure 6  Visualization of the detection results.** Anomalies are marked as yellow circles. Normal points are marked as blue circles. Black lines are the boundaries learned by the methods.

sub-regions on complex datasets to better separate anomaly points from normal points, which is beneficial for the model to gain higher classification accuracy.

## Experiments on UCI datasets

Results on the UCI datasets showed that SA-SVM achieved the best results on all high-dimensional datasets and one low-dimensional dataset, as shown in Table 3. For the three high-dimensional datasets, SA-SVM gained more advantages than its competitors in the ability of anomaly detection. Especially, the dataset U1 has very few anomalies, *i.e.,* anomaly ratio is equal to 1.65%, in this case, the detection accuracy of SA-SVM was still 18.71% higher than that of the competitors. In terms of successfully identifying anomalies, the accuracy of SA-SVM outperformed the three competing methods on the five UCI datasets, which indicates that SA-SVM has a lower risk of misjudging abnormal points than the three competing methods.

The following advantages can be observed from the results of experiments on the synthetic and UCI datasets: (i) The new derived kernel is capable of exploring different sub-regions, which can better separate anomaly instances from normal instances, so as to achieve higher detection accuracy; (ii) Anomaly detection models suffer less negative effects from the complexity of data distribution in the reconstructed feature space than in the background space. Since the space reconstructed by those layered features provides a better spatial environment for anomaly detection.

## Discussion

Compared with the three competitive methods, the proposed SA-SVM showed unique advantages in terms of anomaly detection in high-dimensional space, the detailed interpretation is as following. In Eq. (3), the support vector machine allows the anomalies to be far away from the hyperplane. The slack variables $\xi_i$ in Eq. (3) are used to evaluate

**Peer**J Computer Science

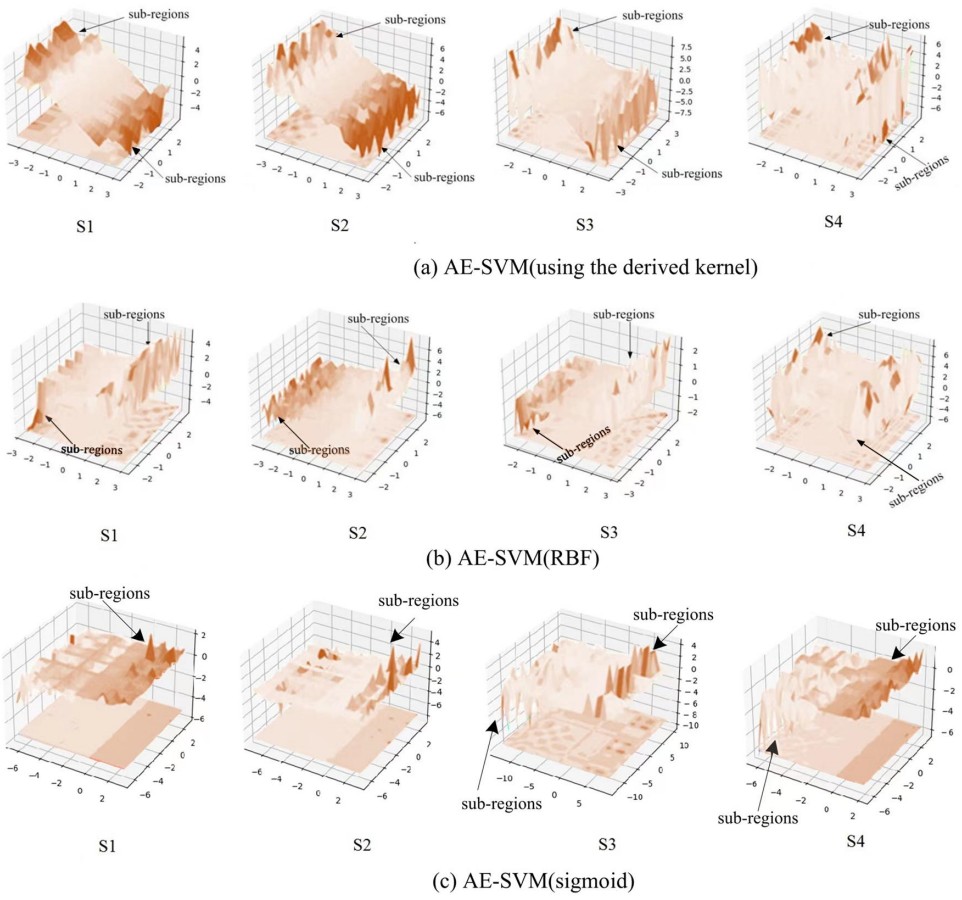

**Figure 7** Visualization of the kernels on the four synthetic datasets.

the errors that anomalies locate outside the hyperplane. Meanwhile, the derived kernel in Eq. (15) achieves separation between anomaly instances and normal instances well. In addition, Eq. (1) extremely reduces the risk of false positive identification. Finally, the objective function in Eq. (16) minimizes variable error, so as to reduce the probability that anomaly instances are misjudged as normal instances. Overall, the proposed SA-SVM is capable of distinguishing anomalies from normal data in high-dimensional space, and can also gain desired anomaly detection precision.

Deep networks possess several layers of nonlinear processing nodes, which can provide a more compact representation of features than non-deep networks, of which autoencoders are typical representations of deep networks. Autoencoders are good at addressing these tasks including capturing latent representations with specific features, feature extraction, data denoising, dimensionality reduction, and compression, etc., for instance, see the examples implemented in *Feng (2019)* and *Zhang et al. (2020)*. Due to the advantages of autoencoders, they are becoming more and more popular in the high-dimensional data reduction dimension.

**Table 3  The detected results on UCI datasets.** The best values are marked in bold.

| | High-dimensional datasets (dimensionality >100 ) | | | Low-dimensional datasets | |
|---|---|---|---|---|---|
| | U1 | U2 | U3 | U4 | U5 |
| SA-SVM | **0.7372 [0.6771]** **{0.7111}** | **0.9937 [0.9388]** **{0.9099}** | **0.8789 [0.8911]** **{0.8661}** | 0.9627 [0.9103] {0.9233} | **0.9962 [0.9775]** **{0.9444}** |
| AE-SVM(RBF) | 0.6222 [0.6001] {0.5733} | 0.9888 [0.9317] {0.8881} | 0.7707 [0.8022] {0.7116} | 0.9000 [0.9004] {0.8662} | 0.9709 [0.9562] {0.9336} |
| AE-SVM (sigmoid) | 0.6312 [0.6229] {0.6060} | 0.9071 [0.9226] {0.8559} | 0.8228 [0.7777] {0.7792} | 0.9447 [0.9009] {0.8988} | 0.9888 [0.9356] {0. 9233} |
| DNN-SVM | 0.7111 [0.6600] {0.5955} | **0.9937** [0.9333] {0.8821} | 0.8719 [0.8772] {0.7944} | 0.9811 [**0.9888**] {0.7944} | 0.9911 [0.9555] **{0.9444}** |
| REPEN | 0.5501 [0.4901] {0.4706} | 0.9911 [0.9309] {0.8871} | 0.8667 [0.8807] {0.8515} | 0.9101 [0.9426] {0.9002} | **0.9962** [0.9423] {0.8991} |
| DevNet | 0.4890 [0.4227] {0.4071} | 0.9237 [0.9306] {0.8995} | 0.7245 [0.8009] {0.8000} | **0.9998** [0.9411] {0.9119} | 0.9790 [0.9599] {0.9075} |
| iForest | 0.5060 [0.4088] {0.3551} | 0.9907 [0.9299] {0.8711} | 0.0822 [0.7807] {0.7112} | 0.6667 [0.7333] {0.6088} | 0.9860 [0.9344] {0.8866} |

Certainly, in addition to deep networks, non-deep networks can also reduce dimensionality of the data, such as PCA (Principal Component Analysis). PCA is not only used for dimension reduction of the data, but also is applied to data visualization (reduce 2-dimension or 3-dimension) and denoising. However, PCA also has some shortcomings, (i) the final dimensionalities of dimension reduction cannot be well estimated. (ii) PCA is mainly to eliminate the correlation between variables, and assumes that the correlation is linear, however, PCA is difficult to obtain good results for dimension reduction of the data having nonlinear dependencies. Indeed, in many applications, the relations between variables are nonlinear, after linear dimension reduction using PCA, the nonlinear correlation between variables may be lost. (iii) PCA needs to assume that the variables obey the Gaussian distribution. When the variables do not obey the Gaussian distribution, such as uniform distribution, scaling and rotation will be occurred.

## CONCLUSION

This article proposed a hybrid method, SA-SVM, which combines a sparse autoencoder with a support vector machine to address the challenge of accurate anomaly detection in high-dimensional space. Experimental results showed that SA-SVM can outperform the state-of-the-art detection methods in terms of anomaly detection performance on both accuracy and F1-score. We demonstrated that the derived kernel is able to explore different sub-regions, which can better separate anomaly instances from normal instances. In addition, our research results suggested that anomaly detection performance can be further improved in the detection space reconstructed by neural networks. In future work, we will look at exploring anomaly detection under the interference of irrelevant attributes in the high-dimensional space, because irrelevant attributes can mask anomalies, resulting in very low anomaly discovery rate.

### Funding

We received funding from the National Natural Science Foundation of China under grant #61871061 to Hongchun Qu, the Sub-project of National Key R&D Program of China under grant #2016YFC500304-01 to DW, and the National key R&D plan ''Research, development and demonstration of key technologies for berry complex ecological cultivation'' 2022YFF1300503-03 to Dianwen Wei. The funders had no role in study design, data collection and analysis, decision to publish, or preparation of the manuscript.

### Grant Disclosures

The following grant information was disclosed by the authors:
National Natural Science Foundation of China: #61871061.
National Key R&D Program of China: #2016YFC500304-01.
National key R&D plan: 2022YFF1300503-03.

### Competing Interests

The authors declare there are no competing interests.

### Author Contributions

- Dianwen Wei conceived and designed the experiments, performed the experiments, prepared figures and/or tables, and approved the final draft.
- Jian Zheng conceived and designed the experiments, performed the experiments, performed the computation work, prepared figures and/or tables, authored or reviewed drafts of the article, and approved the final draft.
- Hongchun Qu analyzed the data, performed the computation work, prepared figures and/or tables, authored or reviewed drafts of the article, and approved the final draft.

### Data Availability

The raw data and code are available in the Supplemental Files.

### Supplemental Information

Supplemental information for this article can be found online at http://dx.doi.org/10.7717/peerj-cs.1214#supplemental-information.

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
