# Peer review of "Anomaly detection for blueberry data using sparse autoencoder-support vector machine"

_PeerJ Computer Science, doi:10.7717/peerj-cs.1214_

## Round 0.1 · original submission · Major Revisions

The paper needs to make the statement clear and improve the writings. Please revise the paper according to the reviewers' comments.

·

Basic reporting

- The paper is written in simple, clear, concise English
- Code and data are available. The code contains one file in which a kernel function is defined as the product of the matérn and squared exponential kernel, plus another file (AESVM), which to me seems to only contain the autoencoder. Thus, the code for the SVM as well as the whole training / evaluation is missing
- It is unclear to me how equation (13) could be derived from (12) and (40), as claimed by the paper. Is this an alternative formulation of the squared exponential kernel that is defined in the code?

Experimental design

The paper addresses the issue of anomaly detection in high dimensional space. However, in my view, it fails to properly formulate the issue fully or address the actual challenge: How to define "outlier" in high dimensions? Any proximity-based method becomes useless in high-dimensional space where data is very sparse. So, fundamentally, a two-stage approach must be taken by (i) projecting the data into low dimensions / finding a low-dimensional manifold, (ii) Detecting outliers in the projected data. Some references are missing that in my opinion are fundamental, such as:

Aggarwal, Charu C., and Philip S. Yu. "Outlier detection for high dimensional data." Proceedings of the 2001 ACM SIGMOD international conference on Management of data. 2001.
Wang, Hongzhi, Mohamed Jaward Bah, and Mohamed Hammad. "Progress in outlier detection techniques: A survey." Ieee Access 7 (2019): 107964-108000.

Validity of the findings

The statement "Meanwhile, the derived kernel in Eq. (15) achieves maximum separation between anomaly instances and normal instances." seems too strong. If at all, the kernel achieves "good" separation, there being no proof of it achieving "maximum" separation. From the presentation, however, it is also unclear to me why said kernel should even achieve "good" separation, as in "better than most other valid kernels".

The statement "In addition, Eq. (16) extremely reduces the risk of false positive identification." refers to equation (16) which does not exist.

The statement "The objective function in Eq. (17) minimizes variable error, so as to reduce the probability of misclassification" seems to refer to a classification problem, whereas the rest of the paper is about outlier detection.

While the results show comparisons to other outlier detection methods, it is unclear why the specific AE architecture and SVM kernel were chosen. I would like to see a suite of experiments with different AE architectures and kernels, and/or a theoretical justification for the chosen ones.

Reviewer 2 ·

Basic reporting

The paper needs through revision. The English and terminology used is not proper in many places. In literature, SA is used to denote simulated annealing whereas the authors in this paper mean something else – sparse autoencoders. More information on data set creation is needed. How is bias prevented in collection or preparation of synthetic data set- needs to be addressed. Background information on auto encoders – one paragraph needs to be added. Some sub-headings are not good: theory, preliminary. Instead say ‘’Background’’ and put everything in that subsection. Details on platform /software used like python or any tools used – no information is given. It should be given. References can be condensed to mot important recently published ones at most 20 or 25.
Page 5 line 20 abnormality detection.
Page 6 line 54 is often of rare? Something is missing.
Line 55 leads to
Line 56 deep methods?
Page 6 Line 58 abnormality searching
Line 59 similarity distance
Line 62 difficulty in measurement.
Line 66 after sparse auto-encoder insert (SA)
Line 67 to capture
Line 71 impose sparse items?
Line 73 delete ‘’used’’
Line 75 there is a need to derive; Cite reference for Mercer Theorem.
Line 82 delete moreover. Warping concave?
Page 7 line 96 Expand OC.
Line 99 Expand LS.
Line 102 Deep neural networks
Line 109 there all needs?
Line 11 similarly
Line 112 Expand GAN
Line 116 REPEN, SVDD
There is too much repetition about use of auto-encoder, its advantages etc. Line 130-136 what is written has appeared few times before.
Line 139 Theorem [34],[35]:
Line 147 define what is the meaning of the symbol o.
Line 151 where is m used?
Line 156 d-dimensional
Line 162 inputting> input
Line 170 implemented on weights?
Define p in (2). In (2) make it (2a) and (2b) since there zre two equations.
Equation (3) the whole argument shall be in brackets min()
Line 178 Dual problem?
Lines 179-180: Cite a reference for Lagrange function.
What is the black dot in (4) and (6)? Define.
Line 190 Cite reference fur KKT conditions.
Line 201 sub-heading can be in italics
Line 202 it can be shown from the Mercer
Line 204 delete ‘’having that’’, in line 214 also
Line 213 and A,B,C are
Line 220 is difficult.
Line 227 ‘’are’’ shall be ‘’being’’
Line 231 delete ‘’having that‘’
Line 233 converges; outputs
‘’as following’’ throughout shall be ‘’as follows:’’
Line 242 ‘’manner’’ shall be ‘’technique’’
Line 251 ‘’it is suitable’’ not ‘’which is suitable’’
Line 259 manner shall be technique. Line 271 also.
Line 260 until shall be after
Line 293 follows [44]
Line 299 expand UCI.
Equation (19) define precision and recall below.
Line 329 fell more slowly?
Line 343 These: which? Expand RBF.
Line 381 representation of features
Line 389 similar? some shortcomings
Lines 399-404 is repeating umpteen times.
Fig.1 You need to use thicker lines to show figure clearly. Font size of letters has be at least 9 or 10.
Captions shall be just one line. All other material shall be in the body of the paper when you refer to the figure.

Experimental design

The tools used etc re missing. That should be included. More on dataset preparation also will be required.

Validity of the findings

Results cannot be verifed by me but I feel that there is some novelty.

Additional comments

The English needs improvememt. Terminology also needs to be changed. s SA shall not be in abbreviated firm.

---

## Round 0.2 · Minor Revisions

The paper is improved. However, there are still a few editing issues. Please make sure to check the paper thoroughly and revise it carefully.

Reviewer 2 ·

Basic reporting

English needs lot of editing: See below:
.The paper is improved but needs editing to improve the English and readability.
Page 2 line 58 Delete rarity in number
Line 123: Mercer Theorem:
Line 164 are the actual
Line 166 item of implementing? Not understandable.
Do not use ‘’having that’’ frequently.
Line 200 it is known from mercer theorem in …
Line 205 non-stationarity and flexibility for controlling
Line 210 are constants
Line 223 vice versa
Line 248 for judging
Line 257 until shall be after? Rewrite the sentence to convey meaning.
Lines 293 and 333 visualized? shows
Lines 323-325 sensitivity
Line 332 what drops? Performance
Line 373 ‘’these’’ shall be ‘’see the ‘’
Line 374 ‘’getting’’ shall be ‘’becoming’’
Line 388 experimental
Line 461 use lower case letters

Experimental design

This is satisfactory. No comment.

Validity of the findings

I find the results correct and I am satisfied.

Additional comments

I need not review again. The author may make all corrections and submit final version to you,

---

## Round 0.3 · accepted · Accept

Congrats to authors for successfully addressing the issues from the reviewers.

Reviewer 2 ·

Basic reporting

The English corrections were implemented as suggested. Now the paper reads well and may be accepted.

Experimental design

No further comments..

Validity of the findings

I believed the conclusions are valid.

Additional comments

None.